# Precision Medicine in Castration-Resistant Prostate Cancer: Advances, Challenges, and the Landscape of PARPi Therapy—A Narrative Review

**DOI:** 10.3390/ijms25042184

**Published:** 2024-02-11

**Authors:** George Dimitrov, Radoslav Mangaldzhiev, Chavdar Slavov, Elenko Popov

**Affiliations:** 1Department of Medical Oncology, Medical University of Sofia, University Hospital “Tsaritsa Yoanna”, 1527 Sofia, Bulgaria; gdimitrov@medfac.mu-sofia.bg (G.D.);; 2Department of Urology, Medical University of Sofia, University Hospital “Tsaritsa Yoanna”, 1527 Sofia, Bulgaria; ch.k.slavov@gmail.com

**Keywords:** PARPi, mCRPC, HRR genes, BRCA1/2 alterations, personalized medicine

## Abstract

After recent approvals, poly-adenosine diphosphate [ADP]-ribose polymerase inhibitors (PARPis) have emerged as a frontline treatment for metastatic castration-resistant prostate cancer (mCRPC). Unlike their restricted use in breast or ovarian cancers, where approval is limited to those with BRCA1/2 alterations, PARPis in mCRPC are applied across a broader spectrum of genetic aberrations. Key findings from the phase III PROPEL trial suggest that PARPis’ accessibility may broaden, even without mandatory testing. An increasing body of evidence underscores the importance of distinct alterations in homologous recombination repair (HRR) genes, revealing unique sensitivities to PARPis. Nonetheless, despite the initial effectiveness of PARPis in treating BRCA-mutated tumors, resistance to therapy is frequently encountered. This review aims to discuss patient stratification based on biomarkers and genetic signatures, offering insights into the nuances of first-line PARPis’ efficacy in the intricate landscape of mCRPC.

## 1. Introduction

Castration-resistant prostate cancer (CRPC), defined by castrate serum testosterone levels (<50 ng/dL or 1.7 nmol/L) coupled with biochemical or radiological progression per the European Association of Urology’s guidelines [1], manifests as a clinically diverse entity marked by heightened mortality [2]. Despite recent therapeutic advances, the landscape of metastatic (m)CRPC treatment has undergone a significant evolution, ushering in a new era of possibilities.

In the current paradigm, first-line treatments for mCRPC primarily center around androgen receptor signaling inhibitors (ARSis) (e.g., abiraterone acetate and enzalutamide) meticulously crafted to thwart the reactivation of the androgen receptor (AR) signaling pathway [3,4]. Simultaneously, taxane-based chemotherapy (docetaxel and cabazitaxel) alongside androgen deprivation therapy [5] serves as a viable strategy [6,7]. Moreover, the benefits of non-drug-based therapeutic modalities, such as immunotherapy with Sipuleucel-T and nuclear medicine approaches exemplified by radium-223, have been demonstrated in this context [8,9]. These strategies aim not only to impede disease progression, but also to enhance overall patient outcomes [10].

Amidst this dynamic landscape, the integration of poly-adenosine diphosphate [ADP]-ribose polymerase inhibitors (PARPis) concurrently with androgen receptor signaling inhibitors emerges as a promising avenue based on the results from several phase III clinical trials [11,12,13,14]. This deliberate combination reflects ongoing efforts to explore synergistic treatment modalities that effectively address the intricate molecular underpinnings of mCRPC [15]. Nevertheless, despite the initial effectiveness of PARPis in treating BRCA-mutated tumors, therapy resistance remains a prevalent challenge, as demonstrated in other solid malignancies [16].

In this narrative review, we delve into the current literature on biomarkers in mCRPC and their implications for prognosis and first-line PARPi treatment response. The ultimate aim is to identify stratification signatures in order to optimize the benefits of this therapeutic approach while minimizing toxicity. This effort contributes to a more personalized and effective treatment paradigm, capitalizing on the concept of synthetic lethality.

## 2. DNA Damage Repair Mechanisms 

Throughout the cell cycle, the cell relies on five fundamental repair pathways to address DNA damage and maintain genetic stability, including mismatch repair, base excision repair, nucleotide excision repair, homologous recombination, and non-homologous end joining [17]. Within the complex process of DNA repair, when alterations affecting or eliminating a single base occur, DNA glycosylases take on a central role. These glycosylases, operating within the base excision repair pathway, remove the modified bases, initiating a cascade of events that lead to the formation of single-stranded DNA breaks. This intricate repair mechanism orchestrates the recruitment of specific endonucleases, with PARP1 playing a pivotal role in coordinating these processes [18]. 

The nucleotide excision repair system is responsible for mending a broad spectrum of single-stranded DNA breaks that disrupt proper DNA coiling. It accomplishes this by excising an oligonucleotide and using the opposite DNA strand as a template for repair [19]. The mismatch repair system specifically targets per base pair mismatches and regulates the homologous recombination repair (HRR) pathway to enhance the efficiency of DNA double-strand breaks’ repair [20]. On the other hand, non-homologous end joining presents an alternative repair process for double-strand breaks that does not depend on an intact template. However, this repair mechanism is less precise than HRR, making it more susceptible to errors, such as deletions and subsequent loss of genetic information [21].

## 3. Poly-Adenosine Diphosphate [ADP]-Ribose Polymerase Inhibitors (PARPis)

Poly (ADP-ribose) polymerase, an essential enzyme in the regulation of DNA repair pathways, holds a pivotal role in cellular mechanisms. PARPis exhibit the ability to trigger synthetic lethality in tumor cells carrying existing mutations in HRR genes such as BRCA1/2, among others [22]. Both olaparib and rucaparib have shown clinical benefits in mCRPC characterized by germline or somatic mutations in HRR genes [23,24]. 

In the PROfound study, the effectiveness of olaparib was assessed in comparison to the physician’s choice of therapy (abiraterone acetate or enzalutamide) for patients with mCRPC who had experienced disease progression while using a new hormonal agent (enzalutamide or abiraterone). Among individuals with at least one alteration in BRCA1, BRCA2, or ATM, olaparib demonstrated superior radiological progression-free survival (PFS), as evidenced by a hazard ratio (HR) for disease progression or mortality of 0.34 (95% CI: 0.25–0.47; *p* < 0.001). However, no statistically significant improvement in overall survival was observed [25]. Conversely, there was no discernible benefit in PFS for the collective cohort of patients possessing any of the 15 predetermined homologous recombination repair (HRR) gene mutations in the PROfound study.

TRITON2, a phase II single-arm study, explored the application of rucaparib in mCRPC characterized by disease progression despite prior treatment with a novel androgen-receptor-targeting agent and taxane-based chemotherapy. Among the patients evaluated and possessing a BRCA1/2 alteration, an objective response rate of 43.5% (95% CI: 31.0–56.7%; 27 out of 62 patients), with a confirmed PSA response rate of 54.8% (95% CI: 45.2–64.1%; 63 out of 115 patients), was reported [24]. 

Expanding on these insights, several clinical trials [PROpel [11], TALAPRO-2 [12], CASPAR [13] and MAGNITUDE [14]] have been designed to evaluate the effectiveness and safety of simultaneous PARPi administration with ADT and androgen receptor signaling inhibitors (ARSis) in the first-line setting for mCRPC (Table 1). Olaparib (PROpel) and rucaparib (CASPAR) inhibit PARP 1-3, while niraparib (MAGNITUDE) and talazoparib (TALAPRO-2) only target PARP 1 and 2. PROpel, CASPAR, and TALAPRO-2 enrolled biomarker-unselected patients, while MAGNITUDE pre-screened subjects for biomarker mutational status. All trials demonstrated benefits in first-line mCRPC for concurrent PARPi and ARSi use, especially in tumors with BRCA2, BRCA1, and ATM pathogenic variants; variations in designs and discordant genetic testing methods contributed to differences in the observed results.

## 4. DNA Repair Genes as Biomarkers

Approximately 30% of individuals diagnosed with mCRPC carry genetic alterations in DNA damage repair-associated genes, specifically those linked to homologous recombination repair (HRR) [15,26,27]. These alterations correlate with adverse clinical outcomes and an accelerated development of resistance to widely employed systemic therapies [26,27,28].

In a study involving 692 males with metastatic prostate cancer, germline DNA samples underwent analysis to detect mutations in 20 DNA repair genes associated with cancer predisposition syndromes [29]. Importantly, this cohort was not selected based on a family history of cancer (including prostate cancer) or the age at diagnosis. The analysis revealed germline mutations in 82 males (11.8%), a significantly higher frequency compared to a cohort of 499 males with localized prostate cancer (2% of males with low- to intermediate-risk cancers and 6% of males with high-risk tumors) or a cohort of 53,105 males without a known cancer diagnosis (2.7%). Notably, mutations were identified in 16 of the 20 genes under study. The predominant gene affected was the effector BRCA2, detected in 37 males (5.3% of those analyzed). Other implicated genes included ATM (11 males, 1.6%), CHEK2 (10 males, 1.9%), BRCA1 (6 males, 0.9%), RAD51D (3 males, 0.4%), and PALB2 (3 males, 0.4%). Additionally, mutations were identified in 11 other DNA repair genes. Notably, the prevalence of these alterations was increased in metastatic cases compared to localized disease [30].

The utilization of somatic (tumor) genomic sequencing approaches to detect DNA repair alterations in prostate cancers has been gaining momentum, aiming to inform therapeutic decisions and facilitate consideration for clinical trials. Collectively, these investigations reveal that germline mutations in HRR genes are present to a lesser extent (in approximately 10% of men with mCRPC), while somatic mutations in these genes are identified more commonly (about 25%) in both primary and metastatic prostate cancer samples [29,31,32]. As a result, the analysis of somatic HRR mutations in CRPC is preferable over germline analysis, whenever feasible. 

Among the HRR genes, BRCA2 variants are the most commonly observed, followed by ATM and BRCA1. The efficacy of PARPis appears to be well established for effector genes of the HRR system (e.g., BRCA1/2, PALB2), while for sensors (e.g., ATM, CDK12, CHEK2) there is marginal or no benefit of PARPi therapy [15]. 

## 5. PARPi Resistance Due to HRR Functional Restoration

Despite the initial efficacy of PARPis in BRCA-mutated tumors, resistance commonly arises, primarily through the restoration of HRR. This restoration, often through BRCA1/2 repair or alterations in the DNA damage response pathway, such as the loss of 53BP1, plays a crucial role in PARPi treatment failure in both preclinical and clinical studies of solid malignancies [33,34]. BRCA reversion mutations, frequently involving deletions of <100 bp, restore the open reading frame of BRCA1/2 and can be detected in circulating cell-free DNA, providing a minimally invasive method to monitor treatment effectiveness and guide further interventions in ovarian cancer [34]. In addition to BRCA1/2, secondary mutations in other HRR genes, like RAD51C and RAD51D, play a role in restoring a functional HRR pathway in ovarian cancer [35]. A study unveiled both primary and secondary mutations in RAD51C and RAD51D. Primary mutations demonstrated sensitivity to PARPis through synthetic lethality, whereas secondary mutations were linked to acquired resistance [35]. In the context of prostate cancer, among the previously mentioned clinical trials, only PROpel assessed the initial status of RAD51C and RAD51D [11], and none of the studies conducted additional testing at treatment failure [12,13,14].

## 6. Additional Stratification for PARPi Resistance

A comprehensive prediction of treatment response is unlikely with a single biomarker. To optimize the effectiveness of PARPi therapy, it is essential to identify additional biomarkers and/or evaluate homologous recombination deficiency (HRD) scores. Using CRISPR-Cas9 technology, it was revealed that modifications in genes encoding the RNase H2 enzyme complex (RNASEH2A, RNASEH2B, and RNASEH2C) induce PARPi sensitivity in cells by impeding ribonucleotide excision repair [36]. Moreover, documented evidence has established a connection between PPP2R2A mutation status and resistance to PARPi therapy [37]. A positive correlation has been observed between HRD scores and improved clinical outcomes in mCRPC patients undergoing PARPi treatment [38]. 

## 7. Current Guideline Recommendations for Genetic Testing in Advanced Prostate Cancer

All guidelines and consensus statements unanimously advocate for comprehensive genetic testing, encompassing both germline and somatic assessments, in men diagnosed with metastatic prostate cancer [10,39]. The National Comprehensive Cancer Network (NCCN) guidelines stand out as the most detailed and comprehensive, spanning prostate cancer, genetic/familial high-risk assessment (breast, ovarian, and pancreatic cancers), and colon cancer [40]. 

Germline testing is strongly recommended for men with high- or very-high-risk prostate cancer, with either regional or metastatic prostate cancer, regardless of family history. Additionally, germline testing is considered essential for men with a personal history of breast cancer or a positive family history of early-onset breast, colorectal, or endometrial cancer (age ≤ 50 years); ovarian, exocrine, or pancreatic cancer (at any age); prostate cancer ≤60 years or prostate cancer death; Lynch syndrome-related cancer, especially if diagnosed before 50 years; or individuals of Ashkenazi Jewish ancestry. Men with prostate cancer may harbor germline mutations in various genes associated with moderate- to high-risk hereditary cancer susceptibility. Notable among these are homologous recombination repair genes such as BRCA2, BRCA1, CHEK2, ATM, PALB2, and RAD51D, mismatch repair genes such as MLH1, MSH2, MSH6, and PMS2, and the pathogenic variant HOXB13.

Somatic testing is advised for men with metastatic hormone-sensitive prostate cancer (mHSPC) or mCRPC. While other major guidelines may provide less specific or comprehensive criteria compared to the NCCN, they all uniformly stress the importance of both germline and somatic testing for men with metastatic prostate cancer, particularly those with a personal or family history or Ashkenazi Jewish ancestry and early-onset disease [10,39].

## 8. Attempts at Overcoming PARPi Resistance

Research has established that platinum or other alkylating agents heighten dependence on PARP enzymes [41]. Although these combinations exhibit efficacy in various solid malignancies, they come at the cost of increased toxicity [42]. PARPis are currently utilized as post-chemotherapy maintenance therapies, as commonly prescribed in advanced ovarian cancer patients, and the clinical feasibility of concurrent administration is still under investigation. The ongoing open-label, single-arm, phase II PLATI–PARP study is actively investigating the simultaneous administration of docetaxel and carboplatin followed by rucaparib in mCRPC patients with alterations in DNA damage repair genes [43]. In a similar setting, mCRPC patients with deleterious mutations in BRCA1/2 and ATM demonstrated no differences in PFS when comparing olaparib to carboplatin treatment [44]. An ongoing phase II trial (NCT04038502), with 100 participants enrolled, is exploring the concurrent administration of olaparib and carboplatin in mCRPC.

While the potential combination of PARPis with other small-molecule inhibitors holds promise, it introduces an elevated risk of toxicity. In the phase II TRAP trial, where mCRPC patients were administered a combination of olaparib and ceralasertib (an ATM inhibitor), a notable benefit in PSA response rates of up to 40% was observed within the subgroup of patients (n = 35) with mutations in homologous recombination repair or the ataxia–telangiectasia gene. However, this benefit came at the cost of increased treatment-related adverse events, with 77% of patients experiencing events of any grade and 21% experiencing grade 3 events, while no grade 4 or 5 adverse events were reported [45]. Likewise, the PI3K-AKT pathway may synergize with PARP, as the inhibition of PARP leads to AKT activation, influenced by its downstream effects on ATM and AKT. In a phase Ib study, the mCRPC cohort exhibited a PSA response in half of the cases with homologous recombination repair, and in about 25% of the cases without HRR [46].

A biological basis for combining immune checkpoint inhibitors concurrently with PARPis has been reported [47]. The inhibition of PARP enzymes induces the release of cytosolic DNA and activates the cyclic GMP-AMP synthase stimulator of interferon genes pathway. This modulation creates a potential environment for the recruitment of immune cells and increased sensitivity of cancer cells to immunotherapy [48]. The exploration of combining PARPis and ICIs was undertaken in the KEYLYNK-010 phase III trial, involving 793 mCRPC patients who had progressed to ARSi and docetaxel and were randomly assigned to pembrolizumab with concurrent olaparib. The trial, at the time of the interim analysis, did not meet its primary endpoints, leading to its discontinuation due to futility [49]. Further investigations into the combination of PARPis and immune checkpoint inhibitors included the phase II CheckMate 9KD trial [50]. In this study, mCRPC patients were concurrently given olaparib with nivolumab. Cohort A1 (n = 88) included individuals previously exposed to chemotherapy and androgen receptor signaling inhibitors, while cohort A2 (n = 77) included chemotherapy-naïve patients or those treated with ARSis. Notably, approximately half of the cases presented homologous recombination repair alterations. While the combination of nivolumab and rucaparib yielded unfavorable results in unselected patients, more favorable response rates and survival outcomes were observed in HRR-positive patients.

The combination of radiopharmaceuticals and concurrent PARPis is also currently being explored in the setting of mCRPC. The COMRADE phase I/II trial delved into the safety and efficacy of combining olaparib with radium-223 in mCRPC patients with bone metastases. The study demonstrated a 6-month rPFS of 58%, with the most pronounced benefit observed in cases with homologous recombination repair gene mutations [51]. 

## 9. Discussion

Despite ADT with ARSis being the primary treatment strategy for a substantial percentage of patients with advanced prostate cancer, resistance to androgen receptor (AR) inhibitors arises in certain cases, particularly those with AR-V7 splice variants, prompting the need for alternative first-line therapeutic approaches [52]. Although PARPis, with or without ARSis, in first-line mCRPC treatment have demonstrated promising results in HRR-mutated patients, challenges in patient selection and stratification arise from variations in study design and genetic testing methodologies. Additionally, combining therapies in oncology often raises concerns about potential increased toxicity and reduced tolerability. Therefore, the successful development of targeted therapy strategies is dependent on establishing reliable assays of response and resistance.

A meta-analysis demonstrated that concurrent PARPis with ARSis in first-line mCRPC yield the highest rPFS advantages across subgroups, particularly pronounced in BRCA1/2-mutated patients. OS benefits are inconclusive regardless of HRR mutation status, with notable increases in grade ≥ 3 TEAEs, especially anemia [53]. Both somatic and germline alterations in BRCA1/2, especially BRCA2 or biallelic mutations, appear to be robust predictors of PARPis’ effectiveness [54]. This may be attributed to the distinct roles of BRCA1 and BRCA2 within the HRR pathway. While BRCA2 is primarily involved in the repair mechanism, BRCA1 plays multiple roles in HR, such as recruitment to DNA damage sites, resection of DNA ends, checkpoint regulation at G2/M and S-phase, and direct participation in DNA repair processes [54]. However, when evaluating the sensitivity and specificity of a biomarker, it is crucial to assess its scope. Limiting the selection to cases with BRCA alterations may capture responsive patients but could potentially overlook others who might have responded if stratified using a broader selection of alterations. Pathological variants in the BRCA1, BRCA2, ATM, ATR, BRIP1, CDK12, CHEK1, CHEK2, DSS1, FANCA, FANCD2, NBSI, PALB2, RAD51B, RAD51C, RAD51D, RAD54, and RPA1 genes are becoming commonly investigated in mCRPC [55]. 

The PROpel trial has faced criticism from the FDA, questioning the demonstration of efficacy and safety beyond the small subgroup of patients with tumor BRCA mutations. Concerns have been raised that the addition of olaparib to abiraterone might be detrimental to patients who are definitively negative for tumor BRCA mutations. As a result, the use of this combination is not recommended by the FDA. However, it has received approval in the European Union, without the need for HRRm testing, when given concurrently with ADT and abiraterone acetate [56].

Reports on PARPis’ administration and treatment outcomes based on genotype are mainly limited to clinical trials. CRPC has significant genomic heterogeneity and demands careful consideration in drug development. Emphasizing the need for molecular stratification in PARPi treatment, current data underscore increased testing for DNA repair defects. Thus, comprehensive assessment of the aggressiveness of newly diagnosed CRPC may not rely on a single biomarker, and there is currently no immunochemical or genetic marker employed to distinguish between different prostate cancer stages [57]. Given the increased toxicity of PARPis and their limited benefits in non-HRR-mutated cohorts, and recognizing that approximately twice the percentage of men diagnosed with mCRPC have somatic DNA repair alterations compared to germline DNA repair defects, a selective recommendation is prudent, especially for HRR-mutated patients. The NCCN guidelines recommend germline mutation testing for all men diagnosed with high-risk disease or metastatic prostate cancer. In the context of mCRPC treatment, priority should be given to testing effector genes such as BRCA2, BRCA1, and those associated with deficient mismatch repair (dMMR). Broader testing, including genes like ATM, PALB2, FANCA, RAD51D, and CHEK2, is currently reserved for eligibility in clinical trials [10]. Consequently, future routine clinical practice should support broader testing for DNA repair defects.

## 10. Conclusions and Future Directions

Genetic testing, as part of personalized medicine, is becoming more and more important in treatment decision-making for men with metastatic prostate cancer, as well as in risk stratification and genetic counseling. Although almost all guidelines recommend genetic workup, the conclusions are vague, with a lack of consensus on the exact protocol, significant heterogeneity, and a lack of good quality evidence. One exception in recent years is the established correlation of BRCA1/BRCA2 genes with increased sensitivity to PARPis in mCRPC. Although mutations in the BRCA1/BRCA2 genes are the most common alterations in mCRPC, regarding the HRR pathway, several other genes are also promising and should be discussed for use as a multi-gene panel in PARPi treatment decision-making. 

The predictive significance of HRR mutations varies across different pathogenic variants, and discrepancies in study design contribute to divergent outcomes in trials assessing first-line PARPis with or without ARSis in mCRPC. Due to heightened toxicity and restricted benefits in non-HRR-mutated cohorts, a judicious and selective recommendation is currently prudent for HRR-mutated patients, particularly emphasizing those with confirmed BRCA2 or biallelic effector mutations, preferably identified through somatic testing.

## Figures and Tables

**Table 1 ijms-25-02184-t001:** A summary of the phase III trials for concurrent use of PARPis in mCRPC.

Trial	Design, Number of Patients Enrolled (n)	Inclusion Criteria	Treatment Arms	Primary Endpoint	HRR Status for Inclusion; Tissue for HRR Evaluation	Key Results
**OLAPARIB**						
PROpel (NCT03732820) [11].	Phase III, randomized (1:1), open-label, n = 796	Patients must be treatment-naïve for ARSis at mCRPC stage; prior taxane for mHSPC was allowed.	Abiraterone (1000 mg once daily) plus prednisone/prednisolone with either full-dose olaparib (300 mg twice daily) or placebo.	Imaging-based progression-free survival (ibPFS) by investigator assessment. Secondary endpoints: OS, TFST, PFS2, and HRQoL.	Somatic or germline HRR mutations in *ATM*, *BRCA1*, *BRCA2*, *BARD1*, *BRIP1*, *CDK12*, *CHEK1*, *CHEK2*, *FANCL*, *PALB2*, *PPP2R2A*, *RAD51B*, *RAD51C*, *RAD51D*, *RAD54L.* Archival tumor sample.	rPFS: from 24.8 to 16.6 months (HR: 0.66, 95% CI: 0.54–0.81, *p* < 0.001) at 19.4 months. HRRm (HR: 0.50, 95% CI: 0.34–0.73) versus non-HRRm (HR: 0.76, 95% CI: 0.60–0.97). OS: 28.6% maturity; hazard ratio, 0.86; 95% CI, 0.66 to 1.12; *p* = 0.29.
**TALAZOPARIB**						
TALAPRO-2 (NCT03395197) [12].	Phase III, randomized (1:1), open-label, n = 805; Cohort 1 (all-comers): non-deficient or unknown n = 636 and HRRm n = 169; Cohort 2: HRRm n = 339.	First-line mCRPC; prior abiraterone acetate or docetaxel for mHSPC was allowed.	Talazoparib 0.5 mg once daily (reduced dose from standard of 1.0 mg) plus enzalutamide 160 mg once daily versus placebo + enzalutamide.	rPFS assessed via blinded independent central review. Secondary endpoints: OS, PRR, TFST, PFS2, and HRQoL.	HRR gene alterations in *BRCA1*, *BRCA2*, *PALB2*, *ATM*, *ATR*, *CHEK2*, *FANCA*, *RAD51C*, *NBN*, *MLH1*, *MRE11A*, *CDK12* using blood samples or the most recent tumor tissue sample.	rPFS: median: not reached versus 22 months; HR: 0.63, 95% CI: 0.51–0.78, *p* < 0.001. HRRm (HR: 0.46, 95% CI: 0.30–0.70, *p* < 0.001) versus non-HRRm (HR: 0.66, 95% CI: 0.49–0.91, *p* = 0.009). OS: Immature, HR: 0.89, 95% CI: 0.69–1.14, *p* = 0.35.
**RUCAPARIB**						
CASPAR (NCT04455750) [13].	Phase III randomized (1:1) n = 984; HRR gene aberration was not required for enrollment.	First-line mCRPC; no prior treatment for mCRPC allowed	Enzalutamide 160 mg once daily and 300 mg rucaparib twice daily or enzalutamide + placebo.	rPFS and OS assessed via blinded independent central review.	*BRCA1*, *BRCA2*, or *PALB2.*	rPFS: HR 0.71 in rPFS (median rPFS of 15 and 21 months in control and combination arms, respectively). OS: HR 0.80 in (median OS of 32 and 40 months, respectively).
**NIRAPARIB**						
MAGNITUDE (NCT03748641) [14].	Phase III, randomized (1:1), open-label, n = 1000; Cohort 1 (HRR+) n = 400; Cohort 2 (HRR−): n = 600.	First-line mCRPC; prior docetaxel for mHSPC and ARSi form nmCRPC or mHSPC was allowed.	Niraparib 200 mg once daily (usual dose: 400 mg) and abiraterone acetate 1000 mg once daily plus prednisone 5 mg twice daily or placebo + abiraterone + prednisone.	rPFS assessed via blinded independent central review. Secondary endpoints: OS, TSP, and TCC.	HRR gene alterations in *ATM*, *BRCA1/2*, *BRIP1*, *CDK12*, *CHEK2*, *FANCA*, *HDAC2*, or *PALB2* using tissue and/or blood samples.	rPFS: 16.5 versus 13.7 months; HR: 0.73, 95% CI: 0.56–0.96, *p* = 0.022 in HRR+ cohort; HRR− cohort (n = 233) demonstrated no benefit with HR = 1.09. OS: Immature, HR: 0.94, 95% CI: 0.65–1.36.

**Abbreviations: mCRPC** metastatic castration-resistant prostate cancer, **HRD** homologous recombination deficiency, **HRR** homologous recombination repair, **ORR** objective response rate, **OS** overall survival, **PRR** PSA response rate: decline of more than 50%, **ibPFS** imaging-based progression-free survival, **rPFS** radiographic progression-free survival, **TSP** time to symptomatic progression, **TCC** time to cytotoxic chemotherapy.

## Data Availability

Not applicable.

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
