# Peer review of "Precision Medicine in Castration-Resistant Prostate Cancer: Advances, Challenges, and the Landscape of PARPi Therapy—A Narrative Review"

_ijms, 2024, doi:10.3390/ijms25042184_

Round 1

Reviewer 1 Report

Comments and Suggestions for Authors

Dear Authors.

The submitted manuscript summarizes the important literature on the use of PARP inhibitors in CRPC. Important studies such as PROfound, PPOpel, TALAPRO-2, CASPAR, MAGNITUDE are reported here.

Information about gene mutations is an important factor examined in these studies and determines the success of the therapy. The authors report on the influence of the most common gene mutations, e.g. BRCA1, ATM or BRCA2, on the effectiveness of PARP inhibitors.

The manuscript is very well written and is an important current topic in the treatment of patients with metastatic prostate cancer. There is also an editorial in the PubMed database that reports on the PROfound, PROpel and TALAPRO studies and comes to the conclusion that more, not less, testing for DNA repair defects should be done (Eur Urol 2023 Sep;84(3):253-256 . doi: 10.1016/j.eururo.2023.03.038. Epub 2023 Apr 20.).

My main criticism of this review by Dimitrov et al. is that there are individual sentences and passages that are not about prostate cancer. This concerns the statements for:

Literature No. 12, 33, 34 and 41. Perhaps the statements should be rephrased so that knowledge of e.g. ovarian cancer also allows these results to apply to prostate cancer? The authors should definitely revise this again.

In addition, the literature cited in line 35 no. 5 and line 62 no. 14 is not suitable as a reference to the statements in the text. References Nos. 28 and 30 are identical.

The reference is missing in line 162.

Please carefully review all literature references again and ensure that they are cited correctly.

Author Response

Esteemed Reviewer,

We appreciate your detailed and professional evaluation of our article. We have carefully considered your recommendations and have made the following modifications, as you have indicated:

  1. The editorial from the PubMed database that reports on the PROfound, PROpel and TALAPRO studies and comes to the conclusion that more, “not less, testing for DNA repair defects should be done” (Eur Urol 2023 Sep;84(3):253-256 . doi: 10.1016/j.eururo.2023.03.038. Epub 2023 Apr 20.) - has been cited and slightly expanded upon.
  2. Individual sentences and passages that are not about prostate cancer. This concerns the statements for: Literature No. 12, 33, 34 and 41. Perhaps the statements should be rephrased so that knowledge of e.g. ovarian cancer also allows these results to apply to prostate cancer? The authors should definitely revise this again. – Literature No. 12, 33, 34 and 41 corrected and clarified as requested.
  3. The literature cited in line 35 no. 5 is not suitable as a reference to the statements in the text.  – Corrected by changing to appropriate reference [Shore, Neal D et al. “Resetting the Bar of Castration Resistance - Understanding Androgen Dynamics in Therapy Resistance and Treatment Choice in Prostate Cancer.” Clinical genitourinary cancer vol. 19,3 (2021): 199-207. doi:10.1016/j.clgc.2020.08.008]
  4. line 62 no. 14 is not suitable as a reference to the statements in the text. - Corrected by changing to appropriate reference [Stracker TH, Osagie OI, Escorcia FE, Citrin DE. Exploiting the DNA Damage Response for Prostate Cancer Therapy. Cancers 2024, 16, 83. https://doi.org/10.3390/cancers16010083].
  5. References Nos. 28 and 30 are identical. – corrected by updating reference 30 with [Hansen, E.B., Karlsson, Q., Merson, S. et al. Impact of germline DNA repair gene variants on prognosis and treatment of men with advanced prostate cancer. Sci Rep 13, 19135 (2023). https://doi.org/10.1038/s41598-023-46323-5]
  6. The reference is missing in line 162. – appropriate references were added as per suggestion.

Reviewer 2 Report

Comments and Suggestions for Authors

This study was reported the role of PARP inhibitor in patients with CRPC who have genetic mutations. The reviewer thinks that this report has useful information for readers. The reviewer would like to suggest some critiques as follows.

1.      On line 32, “next-generation hormonal agents” is inadequate. “ARSI” is better. Furthermore, ARSI is not limited to abiraterone and enzalutamide; other agents can also be used.

2.      On line 35, Sipleucel-T and Radium-223 should be deleted from the text because they have different mechanisms of action.

3.      On line 39, this statement should include a citation.

4.      On line 116, is it true that 30% of CRPC patients have genetic alterations? The citation for this evidence needs to be provided.

5.      On line 144, “while for sensors …   no benefit” is unclear.

6.      In accordance with the submission rules, the manuscript needs to be revised, especially in the References section.

Author Response

Esteemed Reviewer,

We appreciate your detailed and professional evaluation of our article. We have carefully considered your recommendations and have made the following modifications, as you have indicated:

  1. On line 32, “next-generation hormonal agents” is inadequate. “ARSI” is better. Furthermore, ARSI is not limited to abiraterone and enzalutamide; other agents can also be used. – Corrected in main text as per suggestion.
  2. On line 35, Sipleucel-T and Radium-223 should be deleted from the text because they have different mechanisms of action. - The authors agree that, although the article predominantly focuses on the concurrent use of ADT+ARSi+PARPi in the context of mCRPC, it is important to acknowledge the existence of other currently approved therapeutic modalities for this setting. Therefore, the sentence in question was rephrased.
  3. On line 39, this statement should include a citation. – Added relevant citations as per request.
  4. On line 116, is it true that 30% of CRPC patients have genetic alterations? The citation for this evidence needs to be provided. - Added relevant citations as per request.
  5. On line 144, “while for sensors …   no benefit” is unclear. – sentence has been refined.

Round 2

Reviewer 1 Report

Comments and Suggestions for Authors

The authors have corrected the manuscript to my satisfaction. Thank you for the revision. In my opinion, the manuscript is suitable for publication.

Author Response

Dear Reviewer 1, we the authors would like to thank you for you time and positive feedback, as well as for the constructive suggestions, which we believe led to improvement of our manuscript! Best regards!

Reviewer 2 Report

Comments and Suggestions for Authors

ARSi should be used on line 33.

Author Response

ARSi should be used on line 33.

Corrected as per suggestion

Dear Reviewer 2

Thank you for your valuable and insightful feedback on our manuscript. We greatly appreciate your detailed analysis and constructive suggestions.